# Immune Mechanisms of Pulmonary Fibrosis with Bleomycin

**DOI:** 10.3390/ijms24043149

**Published:** 2023-02-05

**Authors:** Yuko Ishida, Yumi Kuninaka, Naofumi Mukaida, Toshikazu Kondo

**Affiliations:** Department of Forensic Medicine, Wakayama Medical University, 811-1 Kimiidera, Wakayama 641-8509, Japan

**Keywords:** pulmonary fibrosis, idiopathic pulmonary fibrosis, bleomycin, inflammation, cytokines, chemokines, growth factors, wound healing

## Abstract

Fibrosis and structural remodeling of the lung tissue can significantly impair lung function, often with fatal consequences. The etiology of pulmonary fibrosis (PF) is diverse and includes different triggers such as allergens, chemicals, radiation, and environmental particles. However, the cause of idiopathic PF (IPF), one of the most common forms of PF, remains unknown. Experimental models have been developed to study the mechanisms of PF, and the murine bleomycin (BLM) model has received the most attention. Epithelial injury, inflammation, epithelial–mesenchymal transition (EMT), myofibroblast activation, and repeated tissue injury are important initiators of fibrosis. In this review, we examined the common mechanisms of lung wound-healing responses after BLM-induced lung injury as well as the pathogenesis of the most common PF. A three-stage model of wound repair involving injury, inflammation, and repair is outlined. Dysregulation of one or more of these three phases has been reported in many cases of PF. We reviewed the literature investigating PF pathogenesis, and the role of cytokines, chemokines, growth factors, and matrix feeding in an animal model of BLM-induced PF.

## 1. Introduction

Idiopathic pulmonary fibrosis (IPF) is a type of idiopathic interstitial pneumonia with high mortality and life expectancy of 2–3 years after diagnosis [1]. The main symptoms of IPF include dyspnea, hypoxemia, prominent lung infiltrates on radiographs, and accumulation of fibroblasts in lung tissue. These symptoms result from the scarring of lung tissue caused by IPF [2].

Although the mechanisms of fibrosis in IPF are not yet fully understood, a general phenomenon in its pathogenesis is a genetic predisposition to subclinical recruitment injuries into the alveolar epithelium, which is followed by pulmonary failure of cyst re-epithelialization and repair [3]. Activated cells within the alveoli release large amounts of cytokines and growth factors that promote pulmonary fibroblast recruitment, proliferation, and differentiation into myofibroblasts, resulting in excessive collagen deposition and progression of the lung parenchyma, causing scars and irreversible loss of function [4,5]. The latest guidelines for treating IPF recommend the use of pirfenidone and nintedanib: two compounds with pleiotropic mechanisms of action [6,7,8]. Unfortunately, both agents have limited efficacy in preventing disease progression and improving the quality of life, and they are associated with tolerance issues [4,5,9,10]. Lung transplantation is the only treatment option for patients with IPF; however, owing to age and comorbidities, it is viable only for a few patients [11]. Therefore, there is an urgent need to develop new therapeutic agents to treat PF.

Although not a single animal model of PF emulates all the features of the human PF, some present several symptoms of IPF. Among the various animal models of PF (BLM, fluorescein isothiocyanate (FITC), silica, radiation, etc.), the BLM model is the most widely used and best-characterized mouse model [12,13]. The BLM-induced PF model is advantageous because it can be easily induced in a short period with high reproducibility. However, a significant drawback of the BLM model is the self-limiting nature of the fibrosis, which is in contrast with the typical progressive chronic fibrosis observed in human IPF [12]. Despite its limitations, the BLM model is appropriate to elucidate the PF pathophysiology and for preclinical research owing to its several advantages and drug discovery capacity.

BLM are a family of complex glycopeptides with antitumor activity; they were originally isolated from the actinomycete *Streptomyces verticillus* [14,15]. Clinically, it is primarily used as an antitumor antibiotic against various cancers, including lymphomas [16,17,18]. BLM toxicity occurs primarily in the lungs, skin, and mucosal organs, and PF is a well-known side effect of BLM [16,19]. This pronounced undesirable side effect of BLM has prompted the studies to understand the mechanisms of PF associated with IPF in animal models.

Fibrogenesis is considered a defective and persistent wound-healing/connective tissue repair in response to repeated alveolar microinjuries. A prominent feature of this fibrotic repair process is the excessive deposition of extracellular matrix (ECM) components, such as hyaluronic acid, fibronectin, and interstitial collagen, that irreversibly restructures the lung tissue, thickening alveolar and bronchial walls and compromising gas exchange [20,21]. In wound healing, fibroblasts are the key cells responsible for ECM synthesis and deposition, as they provide the initial scaffolding for tissue regeneration [20,21]. When abnormal wound healing and fibrosis occur, fibroblasts respond by overproliferating at the site of injury, acquiring a profibrotic phenotype that is resistant to apoptosis and differentiates into contractile myofibroblasts that perpetuate the fibrotic process [20,21]. Activated fibroblasts/myofibroblasts are highly responsive to growth factors and cytokines.

## 2. Animal Models

BLM-treated animals have been reported to have histological features, such as the loss of endobronchial buds, collagen walls, and alveolar space, similar to those seen in patients with IPF [22]. This observation suggests that BLM recapitulates the typical features of human disease, popularizing the use of this model (Table 1). In addition, BLM models are advantageous because they are easy to handle, widely available, reproducible, and meet important criteria for a good animal model [23]. Different fibrotic patterns occur according to the route of administration, and consistent dosages have been established across species to achieve a fibrotic response. Intratracheal administration, the standard route of administration, accentuates central bronchial fibrosis, whereas intravenous or intraperitoneal administration produces subpleural scarring similar to that in human disease [24]. BLM models have significantly contributed to our understanding of the pathophysiological roles of cytokines, growth factors, and signaling pathways involved in PF. For example, transforming growth factor (TGF)-β has been identified as a key factor in the pathogenesis of PF [25].

However, the BLM model has significant limitations when it comes to elucidating the progressive nature of human IPF, despite the similarities in histological changes and their cause. BLM resembles acute lung injury in some respects, as it causes an inflammatory response triggered by the overproduction of free radicals, induction of proinflammatory cytokines, and activation of macrophages and neutrophils. Moreover, the subsequent progression of fibrosis is partially reversible [100]. Therefore, one of the most important features of human IPF is its absence in animal models, which should be considered when using the BLM model.

Several research groups have attempted to model IPF using different species, fibrotic disorders, routes, and methods of administration to induce a fibrotic response in animal lungs. However, there is little consistency across laboratories regarding the optimal protocols for these preclinical models. The BLM model is well characterized, clinically relevant, and allows multiple routes of administration to induce fibrosis. The disadvantage is that the disease may be self-limiting in mice. In the BLM model, fibrosis has been shown to recede spontaneously beyond 28 days [13,23], contrary to the majority of patients, where pulmonary fibrosis does not resolve. Therefore, the use of this model is limited to assess the prophylactic efficacy of potential antifibrotic compounds. Another variable introduced when investigating the potential of a novel antifibrotic compound in an animal model of pulmonary fibrosis is the dosing regimen used for the drug. The administration of compounds that are tested prophylactically is initiated before or on the same day that fibrosis is first induced. On the other hand, when tested therapeutically, this is usually initiated after fibrosis has been established. The current state of preclinical testing can be improved by addressing issues such as the time course of treatment, animal size and characteristics, clinically relevant treatment endpoints, and reproducibility of treatment results. While it is acknowledged that the natural occurrence of pulmonary fibrosis in domestic animals can provide valuable information, rodents are the most essential model for studying the pathogenesis of diseases and to evaluate preclinical treatments. Many traditional and newly developed experimental models provide valuable insights into pathogenesis and help to identify novel therapeutic targets that could be evaluated and validated in clinical trials [23,101,102]. As the role of animal models is to replicate specific aspects of disease, they must be carefully selected, designed, and implemented to bridge the translational gap between benchside and bedside. Currently, the BLM model of pulmonary fibrosis is the cheapest, easiest, fastest, most reproducible, and the most used animal model of IPF. This allows overcoming the poor reproducibility of human disease (Table 2). So far, valuable insights into the pathogenesis, prognosis, and treatment of IPF have been obtained with this model.

## 3. Mechanisms of Wound Healing and Fibrosis

The wound-healing response is often characterized by three distinct phases: injury, inflammation, and repair (Figure 1). Although not all PF cases fit this simple paradigm, it is a useful model for elucidating the general and particular mechanisms of PF.

### 3.1. Phase I, Injury

Injuries from autoimmune and allergic reactions, environmental particulates, infections, and mechanical injuries often disrupt the normal tissue architecture and initiate healing responses. Inflammation after injury also causes cell damage and tissue destruction. Damaged epithelial and endothelial cells must be replaced to maintain barrier function and integrity, respectively, and to prevent blood loss. Acute injury of endothelial cells triggers the release of inflammatory mediators and the initiation of the antifibrinolytic clotting cascade, temporarily blocking injured vessels with platelets and fibrin-rich thrombi [103]. The platelet differentiation factor, X-box binding protein-1, has a higher level of expression in the lung homogenates, epithelial cells, and bronchoalveolar lavage fluid (BALF) from patients with IPF expressed than in those from patients with chronic obstructive pulmonary disease (COPD) patients and controls [104,105], suggesting that clot-forming responses are continuously activated. Thrombin, a serine protease required to convert fibrinogen to fibrin, is also readily detected in the lung and alveolar lumen of some patients with PF and is associated with the activation of the coagulation pathway [106,107,108]. In addition, thrombin directly activates fibroblasts and promotes their proliferation and differentiation into collagen-producing myofibroblasts [109,110]. Damage to the airway epithelium, particularly alveolar lung cells, triggers a similar anti-fibrinolytic cascade, leading to interstitial edema, areas of acute inflammation, and detachment of the epithelium from the basement membrane [108,111].

#### Matrix Metalloproteases (MMPs)

There is increasing evidence that the majority of the MMPs are involved in the pathophysiological mechanisms of IPF. MMP-1, MMP-2, MMP-3, MMP-7, MMP-8, MMP-9, MMP-10, MMP-11, MMP-12, MMP-13, MMP-19, MMP-28, and tissue inhibitors of metalloproteases (TIMPs) exhibit roles in the progression and suppression of IPF [112,113,114]. MMP activity is regulated at multiple levels, including gene transcription, extracellular activation of zymogens, and inactivation by specific TIMPs [115]. Accumulating evidence suggests that an imbalance between MMPs and TIMPs can alter ECM metabolism in diverse lung diseases, including IPF [116,117,118]. Two gelatinases, MMP-2 and MMP-9, are of particular interest because they can degrade type IV collagen and gelatin, which are major components of basement membranes. Cinetto et al. reported that the levels of MMP-2, MMP-9, TIMP-1, and TIMP-2 were increased in the BALF of BLM-treated mice and that glycogen synthase 3 (GSK-3) inhibition may modulate MMP-2, MMP-9, TIMP-1, and TIMP-2 activity in BALF and lung tissue, thus preventing BLM-induced lung injury [93].

Platelet recruitment, degranulation, and thrombus formation rapidly progress to a phase of vasodilation phase with increased permeability, allowing the direct recruitment of leukocytes to the site of extravasation and injury [119]. The basement membrane, which forms the ECM underlying the epithelium and endothelium of the parenchymal tissue, precludes direct access to the damaged tissue. To break down this physical barrier, MMPs cleave one or more ECM components, allowing the cellular access to and from the site of injury. Specifically, MMP-2 and MMP-9 cleave two key components of the basement membrane, namely, N-type collagen and gelatin [120,121,122]. Several studies have reported an upregulation of MMP-2 and MMP-9 in fibrotic diseases, highlighting its role in the processes of tissue destruction and regeneration [123,124,125,126].

Kim et al. reported that MMP-2 and MMP-9 activities in BALF and lung parenchyma initially increased rapidly at the beginning, peaked on day 4, and then continuously decreased [127]. This is in agreement with previous studies that showed an initial increase in MMP-2 and MMP-9 activity followed by a decrease [125,128]. Both MMPs are predominantly expressed in inflammatory cells, more specifically MMP-2 in alveolar macrophages and MMP-9 in neutrophils [127]. Previous studies have revealed that these MMPs co-localize with type IV collagen on the basement membrane, suggesting that the early stages of MMP-2 and MMP-9 secretion by macrophages and neutrophils may play an important role in degrading the basement membrane and promoting inflammatory cell migration [116].

MMP gene-targeted studies in mice have indicated that many MMPs regulate processes involved in the pathogenesis of IPF [113]. MMP-9 levels are increased in IPF BALF samples and they are localized in alveolar and interstitial macrophages, metaplastic airway epithelial cells, and neutrophils in IPF lungs [129,130,131]. Membrane-bound MMP-9 on tumor cells activates latent TGB-β1, but the treatment of wild-type (WT) and Mmp9^−/−^ mice with BLM results in similar lung collagen levels [30,132]. However, MMP-9 activity to regulate the lung fibrosis response is cell-specific; MMP-9 did not affect the lung fibrosis response in transgenic mice overexpressing profibrotic interleukin (IL)-13 in airway club cells [133]. In addition, BLM-treated transgenic mice overexpressing human MMP-9 in macrophages have been reported to experience less severe PF, which is preceded by a significant decrease in the number of neutrophils and lymphocytes in BAL samples and a decrease in TIMP-1 levels in BALF samples [31]. The overexpression of MMP-9 in alveolar macrophages limits the severity of PF induced by BLM in mice by cleaving the insulin-like growth factor (IGF)-binding protein-3 (IGFBP-3), which is a carrier protein that exerts its function through IGFs. However, IGFBP-3 also has IGF-independent effects mediated by binding to TGF-β receptors, and IGFBP-3 is closely related to IPF pathogenesis [134].

Owing to the complexity of MMP expression and regulation, a great deal remains unknown about MMP activity in the PF response to injury. MMPs are expressed in the cells of different tissues and may have beneficial or detrimental effects on different organs. Presently, there are no MMP inhibitors in clinical use for the treating IPF; however, with further basic research, MMP inhibitors can potentially be novel therapeutic agents for IPF.

### 3.2. Phase II, Inflammation

IPF is generally recognized as an inflammatory disease. Damage to the lung tissue causes surrounding tissue inflammation. Inflammatory cells gather and immune cells participate in the tissue repair process. Several chemical factors secreted by these inflammatory cells activate fibroblasts, causing an excessive deposition of ECM components. IPF is also considered to be an epithelium-driven disease [135]. Normal lungs can repair microdamaged type II alveolar epithelial cells [136]. However, when lung microinjury persists, abnormal wound healing activates alveolar epithelial cells, resulting in excessive secretion of mediators, such as TGF-β, platelet-derived growth factor (PDGF), tumor necrosis factor (TNF)-α, angiotensin, and chemokines [137,138]. These mediators promote fibroblast proliferation, migration, and differentiation into myofibroblasts that are resistant to cell apoptosis and secrete ECM components, such as collagen [139,140]. 

Neutrophils, eosinophils, lymphocytes, and macrophages are observed at the sites of acute injury, while phagocytic cells clear cell debris and necrotic areas. The impact of specific inflammatory cells on downstream fibrosis, particularly IPF, remains unknown [141,142,143,144]. One hypothesis stems from the observation that anti-inflammatory agents are largely ineffective in treating patients with IPF and interstitial pneumonia [145,146,147]. However, the timing of inflammatory events may determine the role that inflammatory processes play. Early inflammation, which decreases in the later stages of the disease, may promote wound healing and contribute to fibrosis. As an example, early recruitment of eosinophils, neutrophils, lymphocytes, and macrophages that produce inflammatory cytokines and chemokines may contribute to local TGF-β and IL-13 upregulation [148,149,150,151]. However, after the initial injury and inflammatory response, the late replenishment of inflammatory cells assists phagocytosis, removes cell debris, and controls excessive cell proliferation, which collectively may contribute to normal healing. Thus, delayed inflammation may play an antifibrotic role and could be necessary for terminating the wound-healing response. A late inflammatory profile rich in phagocytic macrophages that help eliminate fibroblasts and regulatory T cells that secrete IL-10 may suppress local chemokine and TGF-β production and prevent excessive fibroblast activation [152,153]. However, it should be recognized that the mechanisms leading to PF are diverse and that numerous genetic, environmental, and immunological interactions regulate the overall process.

#### 3.2.1. Cytokines

The inflammatory response has profound effects on tissue-resident and other inflammatory cells. Inflammatory cells further propagate inflammation by secreting cytokines, chemokines, and growth factors. Many cytokines participate in wound healing and fibrotic responses, and cytokines such as interleukin (IL)-1α, IL-1β, TNF-α, TGF-β, and PDGF are frequently observed in patients with IPF [154]. Of the many cytokines involved in PF, IL-4, IL-13, and TGF-β are of particular interest. These cytokines act through the mobilization, activation, and proliferation of fibroblasts, macrophages, and myofibroblasts, respectively, and they can exhibit significant profibrotic activity [155,156,157,158,159].

IL-1β is considered to play a central role in PF as it can stimulate collagen expression in vitro in a dose-dependent manner [160]. The administration of recombinant mouse IL-1β to wild-type (WT) mice induces severe tissue destruction, with increased inflammation and collagen deposition [36]. Several studies have revealed that uric acid and ATP released from BLM-injured lung cells are the major endogenous danger signals that increase mitochondrial reactive oxygen species (ROS) generation and activate the NALP3 inflammasome, which is characterized by procaspase-1 maturation, leading to IL-1β production and lung fibrosis [94,161,162,163,164,165]. The administration of the caspase-1 inhibitor z-YVAD-fmk or caspase-1 in mice suppresses BLM-induced IL-1β production, lung inflammation, and fibrosis [94]. In addition, IL-1β activates the IL-1R/MyD88 complex in tissue resident cells, particularly in lung epithelial cells, and activates NF-κB. This leads to inflammation, including through neutrophil and lymphocyte recruitment and fibroblast activation [36]. Fluorenidone (FD), a small pyridine drug, suppresses experimental lung inflammation, fibrosis, and protein expression of IL-1β in the BALF of mice [95]. Furthermore, FD inhibits BLM-induced lung inflammation and fibrosis in mice via the activation of the NALP3 inflammasome and inhibition of the IL-1β/IL-1R1/ MyD88/NF-κB pathway [96].

Inflammation associated with BLM lung injury increased the production of the anti-inflammatory cytokine IL-10 by lung macrophages [38], while BLM-treated IL-10^−/−^ mice displayed increased lung inflammation, as reflected by increased lymphocytes and granulocytes in the BALF. However, there was no significant difference in the severity of lung fibrosis between IL-10^−/−^ and WT mice [38], suggesting an immunomodulatory role of IL-10 in the inflammatory response but not in BLM-induced lung fibrosis.

IL-12 participates in the differentiation of naive T cells into Th1 cells [166]. Intratracheally treated *Il12p40*^−/−^ mice with BLM exhibited reduced lung mononuclear cell infiltration but increased lung fibrosis compared with WT controls [39]. Furthermore, the expression of CXCL10, CCL5, and CCL11 was lower in *Il12p40*^−/−^ mice than in the controls. Thus, IL-12 may promote lymphoid tissue response to BLM and suppress PF development.

The Th2 cytokine IL-13 was observed at a high concentration in the blood and BALF of patients with IPF and correlated with disease severity [167]. IL-13 was found to be directly linked to the pathogenesis of PF, and the blocking of IL-13 or deletion of it in the germ line reduced collagen deposition after BLM exposure [168,169,170]. Nie et al. used Akt1^−/−^ mice to modulate BLM-induced lung fibrosis by upregulating the production of the profibrotic cytokine IL-13 in macrophages [97].

Previous studies have shown that IL-17 is associated with profibrotic effects, such as EMT and collagen production, through its interaction with TGF-β signaling [171,172]. The B-cell activating factor is increased in the BALF of patients with IPF; it promotes IL-17 release from Th17 cells and participates in lung fibrosis caused by BLM [98]. IL-27 suppresses PF by inhibiting IL-17 secretion and the JAK/STAT and TGF-β/Smad signaling pathways [41]. In addition, IL-17 production by γδT cells in response to epithelial cell injury is mediated by IL-23 in PF [42].

IL-24 is a member of the IL-20 subfamily of cytokines and is widely expressed in tissues, such as the skin, lungs, and reproductive organs [173]. Current research on IL-24 is primarily focused on cutaneous wound healing and tumor suppression [174,175,176,177]. In *Il24^−/−^* mice, Rao et al. observed protection from BLM-induced lung injury and fibrosis, attenuated TGF-β1 production, and reduced M2 macrophage infiltration [40].

IL-33 is a member of the IL-1 family and functions as an inflammatory alamine when released after stress or cell death [178,179]. Fanny et al. investigated the role of IL-33 in a model of BLM-induced inflammation and fibrosis using mice deficient in the IL-33 receptor (chain suppression of tumorigenicity 2, ST2). The deficiency of ST2 causes acutely exacerbated lung inflammation, and the influx of neutrophils was associated with the increased expression of the chemokine CXCL1 in the lungs. In contrast, M2 macrophages were reduced in the lungs of ST2^−/−^ mice, and lung fibrosis was decreased [43]. These results suggest that acute neutrophilic pneumonia can cause PF associated with the production of M2 polarization in an IL-33-ST2-dependent manner [180].

#### 3.2.2. Chemokines

Experiments in humans and rodents have revealed that CC chemokines play an important role in IPF pathogenesis by recruiting and activating mononuclear cells. CCL2 and CCL3 have been reported to promote macrophage mobilization to the lungs, while CCL17 and CCL22 participate in Th2 cell influx [154]. Furthermore, the CCL3–CCR5 axis has also been proposed to play an important role in regulating the mobilization of fibrocytes and monocytes to fibrotic sites in the lungs [44].

The CCL2-CCR2 axis is a key regulator of monocyte trafficking and plays an important role in inflammatory diseases, including PF [181,182,183]. CCL2 concentration is higher in the BALF and serum of patients with IPF [184]. A study on PF in mice reported that ECM deposition was attenuated in *Ccr2^−/−^* mice and was associated with less macrophage infiltration and macrophage-derived MMP-2 and MMP-9 production [45]. Proteinase-activated receptor-1 (PAR1), a major thrombin receptor, increased CCL2 release. PAR1^−/−^ mice were protected from BLM-induced lung inflammation and fibrosis, and this protection was associated with a significant attenuation of CCL2 induction [99]. In contrast, CCR2^+^ cells may be a potential marker of inflammation during fibrosis [185,186]. Moreover, the mobilization of Ly6C^high^CCR2^+^ inflammatory monocytes from the bone marrow to the lungs in fibrotic lungs is signaled by the elevated CCL2 levels [61]. Importantly, *Ccr2*^−/−^ mice show remarkably reduced lung fibrosis in the BLM model [46,47]. 

BLM treatment enhances the expression of various chemokines in the lungs, including CCL3, in the lungs [49,183,187,188]. In addition, specific receptors for CCL3 and CC chemokine receptors CCR1 and CCR5 are expressed in fibrocytes [189,190]. The passive immunization of BLM-loaded mice with anti-CCL3 antibodies suppresses fibrotic changes [49], suggesting that CCL3 is a mediator of fibrosis induced by BLM. Compared with WT mice, collagen accumulation was suppressed in *Ccl3*^−/−^ and *Ccr5*^−/−^ mice but not in *Ccr1*^−/−^ mice, suggesting that locally produced CCL3 interacts with CCR5 and participates in the mobilization of bone marrow-derived macrophages and fibrocytes, the main producers of TGF-β1 by BLM, and in the subsequent development of PF [44]. In humans, the 32 bp deletion allele in the *Ccr5* gene is more common in Caucasians with a frequency of 0.092 [191,192,193]. As this deletion causes a frameshift and results in a premature stop codon, homozygotes do not express a functional CCR5 protein. Thus, susceptibility to BLM may be determined using CCR5 polymorphisms. Therefore, typing human CCR5 polymorphisms could help to individualize BLM treatment for malignancies.

CCL18 is constitutively expressed at high levels in the lungs and is selectively chemotactic to T cells [194,195,196,197]. The synergistic effects of CCL18 overexpression and BLM injury increased the inflammation in the lungs, especially T-cell infiltration, TNF-α, IFN-γ, MMP-2, and MMP-9 [50]. Despite this synergistic effects on inflammation, CCL18 overexpression attenuated BLM-induced collagen accumulation. CCL18-stimulated T lymphocyte infiltration is profibrotic in healthy lungs, whereas in the inflammatory profibrotic lung setting, it is partially protective against lung fibrosis [50].

Activation of the receptor CXCR2 receptor by CXCL1 and CXCL2 is considered to participate in angiogenesis and the recruitment of neutrophils to inflammatory sites [198,199,200,201,202]. Although neutrophils play an essential role in the host defense against pathogens [203], inflammation and angiogenesis by neutrophils may also contribute to the etiology of various chronic syndromes, including PF [204,205]. DF2162, a CXCR2 antagonist, inhibits IL-8-induced angiogenesis and endothelial cell activation [51].

CXCL6 is enhanced in the BALF of IPF patients. Neutralization with anti-CXCL6 antibody reduced pulmonary neutrophil infiltration, IL-1β, CXCL1, TIMP-1 expression, and collagen deposition in BLM-treated mice [52].

Reportedly, the CXCL12–CXCR4 axis participates in BLM-induced PF. Neutralizing CXCL12 suppresses fibrocyte mobilization and lung collagen deposition [53]. Bone marrow-derived lung CXCR4^+^ cells migrate in response to CXCL12 and differentiate into collagen-producing lung fibroblasts [56]. Similarly, CXCR4 antagonists alleviated BLM-induced PF [54,55].

CX3CL1, a member of the CX3C chemokine family, can bind to its specific receptor, CX3CR1 [206,207]. The lack of CX3CR1 results in reduced macrophage mobilization, fibroblast accumulation, and impaired skin wound healing [208]. The interaction between CX3CL1 and CX3CR1 may contribute to the pathogenesis of lung diseases such as asthma and emphysema [209,210]. Ishida et al. showed that the CX3CL1–CX3CR1 axis regulates fibrocytes and M2 macrophages that can exhibit profibrotic activity and is essential for the development of PF by BLM [57].

#### 3.2.3. Macrophages 

Macrophages play a central role in lung homeostasis and organizing the immune response after injury [58,211,212,213], making them a target for future therapeutic agents. Macrophage abundance in tissues increases after animals are exposed to various lung toxins, including BLM [214,215,216], suggesting that macrophages play a role in acute lung injury. Furthermore, macrophages that accumulate in the lungs early in the course of lung toxic injury are activated to an inflammatory M1 phenotype through the expression of inducible nitric oxide synthase (iNOS) and TNF-α [215,217,218,219]. In the BLM-induced PF model, inflammatory macrophages increased their concentration immediately, peaked on day 3, and gradually decreased until day 21. M1 macrophages are the most abundant in BALF, but after BLM exposure, M2 macrophages gradually increased their concentration, reaching maximum levels on day 14, and correlated with collagen deposition [58,59].

Direct evidence of the role of inflammatory/cytotoxic M1 macrophages in acute lung injury comes from the finding that tissue damage is directly correlated with macrophage functional status. In several experimental models using different approaches, such as pharmacological, genetic, or using macrophage-deficient mice, the suppression or depletion of macrophages improved or prevented lung damage. Blocking the cytotoxic/inflammatory activity of M1 macrophages with anti-inflammatory steroids reduces BLM-induced lung injury [62] and suppresses BLM-induced acute lung injury and iNOS expression in *Ccr4*^−/−^ mice that cannot generate M1 macrophages [48]. CCR2 knockout mice deficient in M1 macrophage migration to injury sites are also protected from BLM-induced oxidative stress and tissue injury [47]. Furthermore, the activation of macrophages exacerbates the tissue damage induced by lung toxins, reinforcing the participation of inflammatory M1 macrophages in the pathogenesis of acute lung injury. The pretreatment of rodents with macrophage activators such as lipopolysaccharides (LPS) or *Bacillus Calmette–Guerin* enhances the acute lung injury induced by endotoxins, radiation, BLM, and ozone [215]. A similar increase in toxicity was observed in response to BLM in mice lacking surfactant protein D, which is a lung collectin that suppresses the inflammatory activity of lung macrophages under homeostasis [63].

The resolution of acute injury and restoration of normal lung structure and function are precisely regulated processes. There is evidence that M2 macrophages stimulate a counter-regulatory mechanism that inhibits the release of inflammatory mediators and activates tissue repair. The levels of anti-inflammatory/wound-repair M2 macrophages are higher in the lungs after exposure to BLM [215]. The emergence of M2 macrophages compared with that of M1 macrophages is delayed, which is consistent with the role of these cells in tissue repair [220]. M2 macrophages promote fibrosis by secreting multiple cytokines that enhance the fibroblast-to-myofibroblast transition and epithelial cell wound healing. Targeting M2 macrophages to suppress the fibrosis-promoting phenotype is a potential therapeutic approach for PF. As discussed previously (in the inflammation section), locally produced CX3CL1 can promote the development of BLM-induced PF, primarily by attracting CX3CR1^+^ M2 macrophages and fibrocytes to the lungs [57]. Li et al. reported that clevudine, approved for treating patients with hepatitis B virus (HBV), can inhibit the profibrotic phenotype (such as CD206 and arginase (Arg)) and promote the antifibrotic phenotype (such as CD86 and IL-10) of M2 macrophages by inhibiting the PI3K/Akt signaling pathway [64]. This effect further reduces myofibroblast activation. EMT by M2 decreases collagen deposition and the secretion of fibrosis-promoting cytokines and restores pulmonary function. Furthermore, RP-832c, which specifically targets CD206 receptors on M2 macrophages, significantly suppresses BLM-induced fibrosis in mouse lungs and downregulates TGF-β1 and α-SMA expression [65]. M2 macrophages produce fibroblast-activating factors and cytokines, such as TGF-β, IL-10, and Arg, which promote myofibroblast proliferation and collagen secretion and are involved in the progression of fibrosis [40,66,67]. Thus, the M2 macrophage marker CD206 may be a potential target for PF treatment. Furthermore, JAK/STAT3 signaling plays a central role in M2 macrophage polarization, suggesting that the JAK/STAT3 signaling pathway participates in tissue fibrosis [68,69].

EMT is a process by which epithelial cells lose polarity, intercellular adhesion, and transition to mesenchymal cells [221]. In patients with PF, the EMT of alveolar epithelial cells results in myofibroblasts [222]. TGF-β activates the Smad signaling pathway and plays an important role in alveolar epithelial cell EMT [223,224]. Zhu et al. reported that M2 macrophages are mobilized in the lungs and that the TGF-β–Smad2 signaling pathway is activated in BLM-induced PF in mice [60]. Furthermore, M2 macrophages secrete large amounts of TGF-β and can induce EMT in lung epithelial cells via the TGF-β–Smad2 signaling pathway in vitro [60].

Macrophage migration inhibitory factor (MIF) inhibition attenuated lung injury and ECM deposition in rats with BLM-induced pulmonary fibrosis [225]. The inhibition of MIF is associated with the reduction in macrophage infiltration in lungs. Furthermore, MIF deficiency suppressed myofibroblast accumulation and fibrosis via the TGF-β1/Smads signaling pathway. Thus, the specific inhibition of MIF is expected to contribute to the treatment of PF.

#### 3.2.4. Dendritic Cells

Dendritic cells (DCs) are potent antigen-presenting cells that are abundant in human fibrotic interstitial lung disease [226,227,228]. However, the role of DCs in the pathogenesis of PF remains unknown. Mature DCs and activated memory T cells accumulate in the lungs of BLM-induced PF mice, while the pharmacological inactivation of DCs attenuates the features of BLM-induced lung injury, suggesting that DCs may maintain lung inflammation and fibrosis in the BLM model fibrosis in BLM models [70]. DCs are composed of heterogeneous populations including myeloid DCs (mDCs), tissue-resident DCs, and plasmacytoid DCs (pDCs). These DC subsets may play distinct roles in different stages of fibrosis. pDC depletion reduced the expression of genes and proteins involved in lung fibrosis, inflammation, and DC differentiation in BLM-treated mice [71].

#### 3.2.5. T Cells

Emerging evidence suggests a regulatory role of Th1/Th2 imbalance in the inflammatory phase of PF [21,229]. Systemic T cell depletion by anti-CD3 antibody attenuates ECM accumulation in a mouse model of BLM-induced PF [72].

Among T lymphocytes, CD4^+^CD25^+^FoxP3^+^ regulatory T cells (Treg) play a pivotal role in maintaining immune system homeostasis, and Treg activity is significantly reduced in patients with IPF [230]; however, conflicting results have been reported regarding the role of Tregs in PF [229,231]. Kotsianidis et al. [230] reported a protective role for Tregs using antibody-based depletion in the fibrosis phase of BLM-induced mouse PF [232]. Furthermore, Kamio et al. demonstrated that splenocytes potently ameliorated PF in vivo using a BLM-induced mouse model of PF [73]. This effect was abolished by the antibody-mediated depletion of Tregs [73].

### 3.3. Phase III, Tissue Repair

Myofibroblast-derived collagen and α-SMA form fibrin scaffolds and an extracellular matrix [233,234]. These structures are collectively referred to as granulation tissue; primary fibroblasts and alveolar macrophages isolated from patients with IPF produce significantly more fibronectin and α-SMA than controls, indicating a state of increased fibroblast activation in patients with IPF [74,235]. Mice lacking the type III domain of fibronectin exhibit significantly less lung fibrosis after BLM treatment than WT mice, suggesting that fibronectin is required for PF development [74,76].

In addition to fibronectin, the ECM is composed of glycoproteins such as PDGF and glycosaminoglycans such as hyaluronic acid, proteoglycans, and elastin [236,237,238,239,240]. Fibroblasts activated by growth factors migrate along the ECM network to repair wounds. Within skin wounds, TGF-β also induces a contractile response and regulates collagen fiber orientation [241]. The differentiation of fibroblasts into myofibroblasts leads to the neoexpression of stress fibers and α-SMA, both of which confer myofibroblasts high contractile activity [242,243]. The extent of ECM bedding and the number of activated myofibroblasts determine the amount of collagen deposited [244]. Thus, the balance of MMPs and TIMPs as well as that of collagen and collagenase shifts from promoting collagen synthesis and increasing collagen deposition to control collagen concentration and prevent it from increasing [117,245,246]. For successful wound healing, this balance is achieved when fibroblasts undergo apoptosis, inflammation begins to end, granulation tissue recedes, and collagen-rich lesions remain. The removal of inflammatory cells, especially α-SMA^+^ myofibroblasts, is essential to end collagen deposition [247]. The signals that initiate fibroblast apoptosis in IPF are poorly understood, although several factors could participate, including cytokine imbalance, genetic causes, and constitutive antiapoptotic pathways similar to those of some cancer cells [118,244,248,249].

#### 3.3.1. Growth Factors 

TGF-β is probably the most well-studied growth factor in fibrosis and is regarded as a prototypic mediator of fibrosis [250]. Of the three isoforms, TGF-β1 has been reported to be primarily involved in PF [251]. It increases the transcription of downstream target genes such as procollagen I and III via the transmembrane receptor serine/threonine kinase and cytosolic Smad2/3 signaling pathways [252]. In particular, the loss of Smad3 ameliorates BLM-induced PF [25]. Secreted modular calcium-binding protein 2 (SMOC2), a factor expressed in almost all tissues, was found to inhibit lung fibrosis progression in SMOC2^−/−^. This was indicated by the decreased levels of TGF-β1, α-SMA, p-Smad2, and p-Smad3 in lung tissue samples [253]. Kurarinone, a prenylated flavonoid isolated from *Sophora Flavescens*, suppresses TGF-β-induced EMT in lung epithelial cells and BLM-induced PF by inhibiting the phosphorylation of Smad2/3 and AKT signaling [77]. Anemarrhenae rhizoma is a traditional Chinese herbal medicine. Shen et al. explored the therapeutic effects of the total extract of anemarrhenae rhizoma (TEAR) on BLM-induced pulmonary fibrosis in rat [254]. TEAR protected rats from fibrosis in a dose-dependent manner, and the antifibrotic activity of TEAR was associated with modulation of the TGF-β1/Smad signaling pathway. In addition, the activation of TGF-β1 production and signaling are considered the cornerstone in the EMT process [255]. Ticagrelor, an antiplatelet, inhibited TGF-β1 production and suppressed Smad3 activation and the signaling pathway in BLM-treated rats [256]. Moreover, ticagrelor suppressed EMT, which was indicated by the increased expression of E-cadherin and decreased expression of vimentin and α-SMA. 

In addition to TGF-β, PDGF is another potent fibrotic growth factor that promotes lung fibrosis via fibroblast activation [257]. The expression of PDGF is increased in epithelial cells and macrophages in the lungs of patients with IPF [258]. Imatinib, a PDGF tyrosine kinase inhibitor, exhibits strong anti-fibrotic effects via the inhibition of mesenchymal cell proliferation in BLM-induced PF [78]. Nintedanib is also a multiple inhibitor of tyrosine kinase receptors that participate in the development of lung fibrosis, including PDGF receptors alpha and beta, VEGF receptors 1, 2, and 3, and FGF receptors 1, 2, and 3 [259]. Further, nintedanib can inhibit the development of lung fibrosis in the BLM mouse model [79].

The main downstream element of TGF-β signaling [260], connective growth factor (CTGF), is a profibrotic growth factor that stimulates fibroblast proliferation and increases ECM deposition [261]. CTGF has been detected in the lung tissue of patients with IPF in both type II alveolar cells and interstitial fibroblasts [262]. Wang et al. reported that the increased expression of α-SMA by recombinant CTGF was reduced in human embryonic lung fibroblasts (HELF) treated with anti-CTGF single chain variable fragment antibody (scFv). Anti-CTGF scFv significantly reduced the number of inflammatory leukocytes in BALF and the hydroxyproline content in the lung tissue of BLM-treated mice [80].

Insulin-like growth factor (IGF-1) is another mediator that may be participate in the pathogenesis of IPF; IGF-1 regulates cell migration and differentiation, and its levels are elevated in the lungs of patients with IPF [263]. Blockade of the IGF-1 pathway using a monoclonal antibody against the IGF-1 receptor (IGF-1R) reduces BLM-induced lung injury [81]. Among the IGF-1R-mediated signaling pathways, the PI3K–AKT–mTOR pathway is fundamental in cell growth and survival [264]. Metformin is one of the most common oral biguanide drugs used to lower blood glucose levels in patients with type 2 diabetes. Metformin suppresses collagen deposition, increases fibronectin and α-SMA levels, and increases IGF-1 and PI3K expression in BLM-treated mice [82]. Somatostatin, also known as somatotropin release inhibitory factor (SRIF), is a member of a family of cyclopeptides produced primarily by normal endocrine cells, gastrointestinal cells, immune cells, neurons, and certain tumors. Lung fibroblasts express somatostatin receptors [83,265]. Somatostatin analogs reduced leukocyte counts and IGF-1 levels in BALF and inhibited PF in mouse BLM [84].

#### 3.3.2. Mesenchymal Stem Cells (MSCs)

Stem cells, particularly MSCs, have several advantages such as migrating into damaged lung tissue, modulation of endothelial and epithelial cell permeability by secreting various paracrine factors, suppressing inflammatory responses, promoting tissue repair, and inhibiting bacterial growth [266]. MSCs are pluripotent stem cells that can differentiate into a variety of cell types, including bone cells (osteoblasts), cartilage cells (chondrocytes), muscle cells (myocytes), and adipocytes, which give rise to bone marrow adipose tissue (fat cells). In addition, MSCs in the stromal and perivascular areas near fibroblasts can regulate myofibroblast activity [267]. These results indicate that MSCs are effective for the treatment of BLM-induced fibrotic lung lesions in animals. Stem cells modulate B cell activity by inhibiting B cell maturation and mobilization to the site of lung injury in IPF. The modulation of B cell activity leads to the suppression of the chronic inflammatory process with the subsequent formation of fibrotic lesions [85]. In a model of BLM-induced lung injury, MSCs reduced the expression of co-stimulatory proteins in DCs and monocyte-derived macrophages, reduced their ability to induce antigen-specific T cell immune responses, and promoted the production of immune cells such as Tregs and cytokines such as IL-10, which have immune regulatory functions [86]. 

Gingivally derived MSCs (GMSCs), which are an abundant source of MSCs and can be harvested at a low cost, were isolated and characterized for the first time in 2009 [268]. Intervention with GMSCs has been reported to reduce BLM-induced PF, inflammation, edema, and apoptosis [87]. The administration of BLM notably increased the gene expression of IL-1β, TNF-α, TGF-β, and MMP-9, and decreased IL-10 expression in lung tissue, but these effects were restored by GMSC intervention [87].

The involvement of Toll-like receptor (TLR) signaling and the therapeutic role of MSCs in MyD88^−/−^ mice were analyzed using human placental-derived MSCs (hfPMSCs) [37]. Compared with WT mice, lung injury in MyD88^−/−^ mice treated with BLM reduced inflammation and fibrosis, along with less TGF-β signaling and ECM production. Thus, hfPMSCs may attenuate BLM-induced lung inflammation and fibrosis, partly through the MyD88-mediated attenuation of inflammation in mice.

#### 3.3.3. Fibrocytes

Traditionally, fibroblasts have been considered mesenchymal tissue-derived/endogenous cells. However, recent research has established the concept of circulating bone marrow-derived cells called fibrocytes that migrate into tissues and differentiate into fibroblasts and myofibroblasts [269]. Fibroblasts express myeloid markers, such as CD45 and CD34, chemokine receptor CXCR4, and collagen-I [199,269,270,271,272]. Fibrocytes produce ECM components (collagen I, collagen III, fibronectin, and vimentin), cross-linking enzymes (lysyl oxidase family), cytokines (TNF-α, IL-6, IL-8, and IL-10), chemokines (CCL2, CCL3, CCL4, and CXCL4), growth factors (VEGF, PDGF, GM-CSF, and others), and various MMPs [273,274,275]. Moeller et al. discovered that circulating fibrocytes are increased in patients with IPF and are a prognostic marker and predictor of early death [276]. CXCL12 participates in the mobilization of circulating fibrocytes at the site of lung injury [53]. The neuronal guidance protein slit guidance ligand 2 secreted by fibroblasts suppresses fibrocyte differentiation and BLM-induced lung fibrosis [88].

#### 3.3.4. Fibroblasts and Myofibroblasts

IPF is characterized by fibroblast accumulation, collagen deposition, and ECM remodeling [277,278]. A large amount of myofibroblasts is found in lung tissues from patients with PF, and myofibroblasts are believed to be key effector cells responsible for matrix deposition and structural remodeling [279,280]. Myofibroblasts arise from circulating progenitors, resident fibroblasts, and from alveolar epithelial cells that have undergone EMT, which can be induced by TGF-β [280,281]. In addition, M2 macrophages secrete large amounts of TGF-β. Zhu et al. have demonstrated that M2 macrophages induce EMT through the TGF-β/Smad2 signaling pathway [60].

In humans with IPF and in mouse model of lung fibrosis, adenosine monophosphate (AMP)-activated protein kinase (AMPK) activity is lower in fibrotic regions associated with metabolically active and apoptosis-resistant myofibroblasts [282]. Metformin is a safe and widely used agent for non-insulin-dependent diabetes, and it has therapeutic potential to restore glucose and lipid metabolic homeostasis [283]. Further analysis has revealed a deficient AMPK activation in non-resolving, pathologic fibrotic processes. This supports the role of metformin in reversing the established fibrosis by facilitating deactivation and apoptosis of myofibroblasts [282]. In addition, nintedanib is a potent small molecule inhibitor of the receptor tyrosine kinases PDGF receptor, FGF, and VEGF receptors. Nintedanib has shown consistent antifibrotic activity by reducing fibroblast proliferation, migration and differentiation, and the secretion of ECM, in animal models of lung fibrosis [284]. Therefore, these data provide a strong rationale for the clinical efficacy of nintedanib in patients with IPF.

Previous studies have reported that the intermediate filament protein nestin plays an important role in tissue regeneration and wound healing in various organs [285,286]. Wang et al. found that nestin expression was primarily localized to lung myofibroblasts and increased in a mouse model of pulmonary fibrosis and in patients with IPF [287]. The deficiency of nestin significantly alleviated pulmonary fibrosis in a mouse model. Therefore, targeting nestin expression in lung myofibroblasts may alleviate lung fibrosis and is a promising therapeutic strategy for IPF. On the other hand, tripartite motif-containing 33 (TRIM33) is an E3 ubiquitin ligase known as a negative regulator of TGF-β1 signaling through Smad4 ubiquitination and turnover of TGF-β receptors [288,289]. TRIM33 was overexpressed in alveolar fibroblasts in IPF patients and lungs of BLM-treated mice [290]. In primary lung fibroblasts, *Trim33* deficiency increased the expression of genes downstream of TGF-β1. Since TRIM33 is a negative regulator of fibrosis, the potential therapeutic effect of IPF is expected.

In a rat model of BLM-induced PF, baicalin, the biologically active form of the dried roots of *Scutellaria baicalensis* Georgi, suppressed the severity of PF by reducing the amount of hydroxyproline in lung tissue [291]. Furthermore, BLM promoted fibroblast viability in vivo in a dose-dependent manner, which was limited after treatment with baicalin. These findings suggest that baicalin exerts an inhibitory effect on PF and fibroblast proliferation by BLM.

#### 3.3.5. Induced Pluripotent Stem (iPS) Cells

iPS cells are cells primed by transferring specific pluripotency genes from adult somatic cells, paving the way for new stem cell-based therapies [292]. Zhou et al. demonstrated that by suppressing the TGF-β1-Smad2/3 signaling pathway, EMT, and inflammatory responses, iPS cells have a BLM-induced therapeutic effect in PF mice [89].

#### 3.3.6. Exosomes

Exosomes are extracellular vesicles (EVs), and they are an important point of discussion in the field of EV research. They are nanoscale vesicles secreted by cells under physiological or pathological conditions [293,294,295,296]. The function of exosomes as important mediators of intercellular signaling depends on their loading profiles. Loading profiles include miRNAs, mRNAs, DNA, and proteins that contribute to both the physiological functions and the pathophysiology of various diseases, including PF [293,297,298]. Exosomes mediate intracellular communication between macrophages and fibroblasts and are thought to play an important role in BLM-induced PF. Sun et al. observed that exosomes with macrophage-derived angiotensin II type 1 receptor (AT1R) activate the renin–angiotensin system (RAS) and the TGF-β-Smad2/3 pathway through a shift to the Ang II–AT1R axis, promoting collagen synthesis and mediating BLM-induced lung fibrosis [90]. In addition, a single intravenous administration of purified exosomes derived from human bone marrow mesenchymal stem cells (BM MSC) effectively prevented and restored the core features of BLM-induced PF, improved lung morphology, blunted collagen deposition, and restored lung structure [91].

Exosome-mediated intercellular communication is essential for activating fibrosis-promoting pathways in IPF [299,300]. Similarly, evidence suggests increased exosome synthesis in patients with IPF [92,301]. Martin-Medina et al. discovered more EVs in BALF and BLM-challenged mice with IPF compared to healthy controls. They also reported an increase in WNT5a, a protein associated with fibroblast proliferation and activation, in EVs obtained from primary human fibrotic lung fibroblasts (HFLF) and fibroblasts activated with TGF-β [92]. 

## 4. Conclusions and Future Directions

In this review, we discussed the use of BLM models of PF for describing disease states in vivo and identifying novel drug targets, and we also addressed their use in demonstrating the efficacy of potential compounds. However, BLM models may have limited use for the detailed evaluation and assessment of novel agents for clinical use. The degree of inflammation and ECM deposition contribute to the net deposition of collagen, which ultimately determines whether a fibrotic lesion develops. Therapeutic interventions that inhibit fibroblast activation, proliferation, and apoptosis require an understanding of all stages of wound repair. These three stages of wound repair often appear sequentially; however, in chronic or repetitive injury, these processes occur in parallel, placing a heavy burden on regulatory mechanisms. In humans with IPF and mouse BLM models of PF, inflammation is considered to play a role in disease onset and progression. However, the underlying mechanisms remain unclear. One abnormal response propagates into other mechanisms, and the well-controlled healing response gradually transforms into a fibrotic lesion. In PF, the onset and progression of the healing response becomes uncontrolled, and many delicate balances are broken. Despite significant advances in our understanding of these pathways, therapeutic interventions and new treatments for PF are lacking. Because PF has numerous causes, forms, and stages, the heterogeneity of the disease must be considered when evaluating the results from individual mouse models, and most important, when designing and implementing new therapeutic strategies. Lung wound repair is a highly dynamic process in which immunology, structural biology, and physiology participate. Cooperation between these systems is essential for successful repair.

Recently, single-cell RNA sequencing (scRNA-seq) has revealed the potential to overcome the large-scale changes and spatial heterogeneity in cell types and allowed the reliable identification of related cell populations and confirmation of the complex molecular procedures in IPF. For example, scRNA-seq data provided insight into the heterogeneity and plasticity of endothelial cells (ECs) in normal and fibrotic lungs and suggested that the potential cross-talk between ECs, macrophages, and stromal cells contributes to pathologic IPF [302]. Adipose-derived mesenchymal stem cells (ADSCs) have been reported to exert therapeutic effects in PF, but the underlying mechanisms remain unclear. Rahman et al. injected ADSCs intratracheally into BLM-treated mice and performed scRNA-seq with lung-derived cells. ADSC administration dramatically altered the transcriptome profile and composition of lung macrophages and reduced lung inflammation and fibrosis [303]. Moreover, Aran et al. have identified a profibrotic macrophage subpopulation that localizes to sites of fibrotic scar with activating effects on the mesenchyme [304]. Using scRNA-seq techniques with isolated mouse lung fibroblasts, it was observed that although fibroblast numbers did not increase after BLM treatment, different populations of activated fibroblasts could be identified in fibrotic mouse lungs [305]. In addition, to comprehensively classify lung fibroblast populations with a nonbiased approach, Xie et al. performed scRNA-seq on mesenchymal preparations of lungs from intact or BLM-treated mice. Single-cell transcriptome analysis classified and defined six types of mesenchymal cells in normal lungs and seven types in fibrotic lungs [306]. The collection of these single-cell transcriptomes and the clear classification of the various cell subsets will provide a new resource for understanding the various cellular landscapes and their role in fibrotic disease.

Although treatments for pulmonary fibrosis have been developed in recent years, the prognosis is still discouraging, and there is a need to develop new therapeutic agents. When a receptor activator of NF-κB ligand (RANKL) partial peptide, MHP1-AcN, was administrated in the BLM-induced lung fibrosis model, less collagen deposition was observed [307]. Mansour et al. investigated the antifibrotic potential of tadalafil for BLM-induced pulmonary fibrosis. Tadalafil inhibited TGF-β production and collagen deposition [308]. In addition, sodium houttuyfonate (SH) could attenuate BLM-induced lung injury by reducing the inflammation in fibrogenesis [309]. DRDE-30, an analog of amifostine, suppressed BLM-induced increases in BALF TGF-β and pulmonary hydroxyproline levels while decreasing α-SMA expression, suggesting the suppression of EMT [310]. Moreover, irbesartan, an angiotensin receptor blocker (ARB) with peroxisome proliferator-activated receptor (PPAR)γ activity, reduced lung hydroxyproline levels, BALF leukocyte counts, TGF-β1, and CCL2 levels [311]. These studies indicate that these treatment options could ameliorate pulmonary fibrosis caused by BLM, reinforcing their potential use as adjuvants to alleviate the side effects of BLM [312].

## Figures and Tables

**Figure 1 ijms-24-03149-f001:**
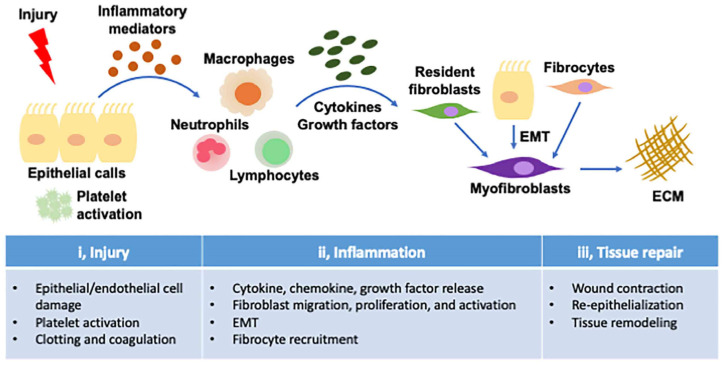
Mechanism of wound healing and pathophysiology of pulmonary fibrosis. The three-step model of injury and wound healing describes the distinct stages of a successful response to injury. After lung injury, epithelial cells release inflammatory mediators and initiate an antifibrinolytic coagulation cascade that triggers platelet activation and thrombus formation. Leukocyte infiltration then begins. Activated macrophages and neutrophils eliminate dead cells and invading organisms. Subsequently, bone marrow-derived fibroblasts and intrinsic fibroblasts proliferate and differentiate into myofibroblasts, which release extracellular matrix (ECM) components. Fibroblasts and myofibroblasts can also be derived from epithelial cells that have undergone epithelial–mesenchymal transition (EMT). In the final stage, activated myofibroblasts can promote wound repair, resulting in wound contraction and vascular recovery. However, if any stage of the tissue repair program becomes out of control or if lung-damaging stimuli persist, fibrosis often develops.

**Table 1 ijms-24-03149-t001:** Studies of BLM-induced PF model focused on immune mechanism.

Approach	PF vs. Controls	Description	References
*Mmp3* ^−/−^	↓	MMP-3 induced EMT through activation of β-catenin signaling.	[26]
*Mmp7* ^−/−^	↓	Decreased pulmonary inflammation and fibrosis.	[27]
*Mmp8* ^−/−^	↓	Increased lung inflammation, but reduced fibrosis by enhancing lung levels of IP-10, MIP-1α, and IL-10.	[28,29]
*Mmp9* ^−/−^	No change	Minimal alveolar bronchiolization.	[30]
Transgenic mice overexpressing MMP-9 in alveolar macrophages	↓	Reduced neutrophils and lymphocytes in BAL. Antifibrotic effect via IGFBP3 degradation.	[31]
*Mmp12* ^−/−^	No change	Similar number of pulmonary macrophages and levels of MMP-2 and MMP-9 as in WT mice.	[32]
*Mmp13* ^−/−^	↑	Exaggerated pulmonary inflammation and fibrosis.	[33]
*Mmp19* ^−/−^	↑	Antifibrotic effect through decreased PTGS2.	[34]
*Mmp28* ^−/−^	↓	Deficits in M2 polarization.	[35]
IL-1R1^−/−^ and MyD88^−/−^	↓	Blockade of IL-1R1 by IL-1 receptor antagonist also reduced inflammation and fibrosis. BLM-induced PF required the inflammasome and IL-1R1/MyD88 signaling.	[36]
MyD88^−/−^	↓	Reduced lung inflammation and fibrosis by reducing TGF-β signaling. Human placental MSCs of fetal origin (hfPMSCs) also attenuated fibrosis with reduced MyD88 and TGF-β signaling activation.	[37]
IL-10^−/−^	No change	Increased inflammatory cells in BALF.	[38]
IL-12p40^−/−^	↑	Less pulmonary mononuclear cell inflammation, whereas increased PF.	[39]
IL-24^−/−^	↓	Attenuated M2 macrophage recruitment and TGF-β1 production.	[40]
IL-27 treatment	↓	Alleviate pulmonary fibrosis by regulating Th17 differentiation.	[41]
IL-1R1^−/−^, IL-23p19^−/−^, and IL-17RA^−/−^	↓	IL-1β-IL-23-IL-17 axis is essential for pulmonary inflammation and fibrosis.	[42]
IL-24^−/−^	↓	Attenuated TGF-β1 production and reduced IL-4-induced production of M2 macrophages.	[40]
IL-33 receptor ST2^−/−^	↓	Augmented acute inflammation, but decreased lung fibrosis with reduced M2 macrophages.	[43]
CCL3^−/−^ and CCR5^−/−^	↓	Collagen accumulation was reduced in CCL3^−/−^ and CCR5^−/−^ but not CCR1^−/−^ mice. The CCL3–CCR5 axis was involved in the accumulation of macrophages and fibrocytes.	[44]
CCR2^−/−^	↓	Fewer macrophages in BALF and macrophage-derived MMP-2 and MMP-9 production. Reduced TNF-α and TGF-β1.	[45,46,47]
CCR4^−/−^	↓	CCL17-dependent activation of CCR4^+^ macrophages played a central role in free radical-induced pulmonary injury.	[48]
Anti-MIP-1α antibody	↓	Reduced pulmonary mononuclear phagocyte accumulation and fibrosis.	[49]
CCL18 overexpression	↓	Exacerbated inflammation but attenuated collagen accumulation.	[50]
CXCR2 antagonist, DF2162	↓	Attenuated collagen deposition with reduced TGF-β1, IL-13, and IL-8.	[51]
anti-CXCL6 antibody	↓	Reduced pulmonary neutrophil influx and collagen deposition.	[52]
anti-CXCL12 antibody	↓	Attenuated recruitment of CD45^+^Col I^+^CXCR4^+^ fibrocytes and attenuated fibrosis.	[53]
CXCR4 antagonist, TN14003 or AMD3100	↓	Inhibited the migration of human fibrocytes in response to CXCL12 in vitro and reduced the trafficking of fibrocytes into the murine lungs in vivo.	[54,55]
Recombinant CXCL12 and CCL21	N.A.	Fibrocytes expressed CXCR4 and CCR7 and could chemotactically respond to both CXCL12 and CCL21.	[56]
CX3CR1^−/−^	↓	Similar levels of inflammation but reduced fibrosis. CX3CR1^+^ macrophages displayed profibrotic M2 phenotypes. Fibrocytes expressed CX3CR1.	[57]
Blocking pulmonary macrophage infiltration	↓	Suppression of the Wnt/β-catenin signaling pathway could attenuate myofibroblast differentiation of lung resident mesenchymal stem cells induced by M2 macrophages and fibrosis.	[58]
M2 macrophages	N.A.	F4/80^+^CD11c^+^CD206^+^ M2 alveolar macrophages peaked on day 14 and were correlated with the magnitude of fibrosis.	[59]
M2 macrophages	↑	M2 macrophages induced EMT through the TGF-β/Smad2 signaling pathway.	[60]
Depletion of lung macrophages	↓	Depletion of Ly6C^hi^ circulating monocytes also reduced PF.	[61]
Dexamethasone treatment	↓	Reduced the inflammation, lung damage, and fibrosis.	[62]
Surfactant protein D (SP-D)^−/−^ or SP-D-overexpression	SP-D^−/−^, ↑; SP-D-overexpression, ↓	SP-D had an anti-inflammatory role via modulation of oxidative–nitrative stress.	[63]
Clevudine treatment	↓	Suppressed profibrotic phenotype of M2 macrophages by inhibiting PI3K/Akt signaling pathway.	[64]
Targeting CD206 receptor, RP-832c	↓	Decreased expression of inflammatory cytokines and TGF-β1 and subsequent fibrosis.	[65]
Effective-compound combination (ECC)	↓	Reduced M2 polarization through the promotion of autophagy via mTOR signaling suppression.	[66]
C/EBP homologous protein (Chop)^−/−^	↓	Attenuated M2 macrophage recruitment and TGF-β1 production.	[67]
p-JAK2/p-STAT3 inhibitor, JSI-124	↓	JSI-124 could modulate fibrosis, autophagy, senescence, and anti-apoptosis.	[68]
A macrolide antibiotic, tacrolimus	↓	Inhibited JAK2/STAT3 signaling in macrophages and reduced the release of profibrotic factors secreted by M2 macrophages.	[69]
DC inhibitor, VAG539	↓	Decreased inflammation and fibrosis.	[70]
pDC depletion, anti-PDCA-1	↓	Reduces immune cell infiltration and reduces the expression of genes and proteins involved in chemotaxis, inflammation and fibrosis.	[71]
Anti-CD3 antibody	↓	Suppressed IL-2 and IL-4 production by lung lymphocytes.	[72]
Adoptive Treg transfer	↓	Suppressed FGF9 and CCL2 production. IL-10^−/−^ abolished the ameliorative effect of Tregs on PF.	[73]
Extra type III domain A (EDA)^−/−^	↓	Decreased TGF-β expression and fibrosis. EDA-containing fibronectin played a critical role in fibrogenesis.	[74]
Fibronectin	N.A.	Fibronectin accumulates during the early inflammatory phase, parallelling hyaluronan accumulation and preceding the development of PF.	[75,76]
Smad3^−/−^	↓	Suppressed type I procollagen gene expression and reduced collagen deposition.	[25]
A prenylated flavonoid, kurarinone	↓	Suppressed the TGF-β-induced EMT via suppressed phosphorylation of Smad2/3 and AKT.	[77]
A tyrosine kinase inhibitor, imatinib	↓	Suppressed fibrosis by reducing the number of mesenchymal cells incorporating bromodeoxyuridine.	[78]
A small molecule inhibitor targeting the receptor kinases of PDGF, bFGF and VEGF, BIBF 1000	↓	Reduced collagen deposition and profibrotic gene expression.	[79]
Anti-CTGF scFv	↓	Reduced the numbers of inflammatory leukocytes in BALF and the hydroxyproline content of lung tissue.	[80]
Anti-IGF-I receptor antibody, A12	↓	Increased fibroblast apoptosis and subsequent resolution of PF. IGF-I increased lung fibroblast migration via the insulin receptor substrate-2/phosphatidylinositol 3-kinase/Akt axis.	[81]
Antidiabetic agent, Metformin	↓	Inhibited enhanced IGF-1 and PI3K expression.	[82]
A somatostatin analogue, SOM230 (pasireotide)	↓	Reduced BAL inflammatory cell influx and expression of CTGF and TGF-β.	[83]
Somatotropin-release inhibiting factors (SRIFs), somatostatins	↓	Reduced the number of neutrophils and lymphocytes, and IGF-1 level in BALF.	[84]
CD19^−/−^ and CD19 transgenic mice	CD19^−/−^, ↓; CD19 Tg, ↑	The levels of IL-6 and immunoglobulin in BALF correlated with CD19 expression levels and the severity of PF.	[85]
Amniotic mesenchymal stromal cells (hAMSCs) injection	↓	hAMSCs reduced pulmonary B-cell recruitment, retention, and maturation, and they counteract the formation and expansion of intrapulmonary lymphoid aggregates.	[86]
Gingival-derived mesenchymal stem cells (GMSCs) administration	↓	Reduced neutrophil elastase, MMP-9, lysophosphatidic acid, lysophosphatidic acid receptor 1, and TGF-β release.	[87]
Recombinant Slit2	↓	Fibroblast-derived Slit2 inhibits fibrocyte differentiation.	[88]
iPS cells	↓	Repressed the expression ratios of MMP-2/TIMP-2 and MMP-9/TIMP-1. Inhibited activation of TGF-β1-Smad2/3 signaling and EMT.	[89]
An exosomes inhibitor, GW4869	↓	Macrophage-derived exosomes containing the angiotensin II type 1 receptor (AT1R) had profibrotic effects by upregulating the Ang II/AT1R axis in fibroblasts.	[90]
MSC-derived exosomes (MEx)	↓	Transplantation of BM-derived monocytes that were preconditioned by MEx prevented collagen deposition, restored lung architecture.	[91]
Extracellular vesicles (EVs)	N.A.	EVs was found in PF, carry fibrotic mediator such as WNT5A, and contribute to fibrogenesis.	[92]
GSK-3 inhibition	↑	Decreased MMP-2 and MMP-9, and enhanced ECM deposition.	[93]
Uric acid release	↑	Constitutes an endogenous danger signal that activates the NALP3 inflammasome, leading to IL-1β production, and fibrosis.	[94]
Fluorofenidone	↓	Antifibrotic effect include regulating caveolin 1 expression and blocking MAPK signaling pathways and/or the IL-1β/IL-1R1/MyD88/NF-κB pathway.	[95,96]
Akt1^−/−^	↓	Decreased production of profibrotic cytokine, IL-13, by macrophages.	[97]
Genetic ablation or neutralization of B-cell activating factor (BAFF)	↓	Reduced the production of TGF-β1 and collagen deposition.	[98]
proteinase-activated receptor-1 (PAR1)^−/−^	↓	Reduced inflammation and fibrosis with attenuation in CCL2 induction.	[99]

↓: alleviation of pulmonary fibrosis (PF) vs. controls, ↑: aggravation of PF vs. controls, N.A.: not applicable. Abbreviations: Mmp, matrix metalloproteinase; EMT, epithelial–mesenchymal transition; IL, interleukin; MIP-1α, macrophage inflammatory protein-1α; BAL, bronchoalveolar lavage; IGFBP3, insulin-like growth factor binding protein 3; WT, wild type; PTGS2, prostaglandin–endoperoxide synthase 2; MyD88, myeloid differentiation factor 88; IL-1R1, IL-1 receptor 1; TGF-β, transforming growth factor-β; MSC, mesenchymal stem cell; hfPMSC, human placental MSCs of fetal origin; BALF, BAL fluid; PF, pulmonary fibrosis; IL-17RA, IL-17 receptor; ST2, chain suppression of tumorigenicity 2; CCL, CC chemokine ligand; CCR, CC chemokine receptor; TNF, tumor necrosis factor; CXCR, CXC chemokine receptor; CXCL, CXC chemokine ligand; Col I, collagen type I; CX3CR, CX3C chemokine receptor; PI3K, phosphatidylinositol 3-kinase; mTOR, mammalian target of rapamycin; JAK, Janus kinase; p-STAT, phosphorylated-signal transducer and activator of transcription; DC, dendritic cell; FGF, fibroblast growth factor; Treg, regulatory T-cell; PDGF, platelet-derived growth factor; bFGF, basic fibroblast growth factor; VEGF, vascular endothelial growth factor; CTGF, connective tissue growth factor; scFv, single-chain variable fragment antibody; IGF-I, insulin-like growth factor I; iPS cell, induced pluripotent stem cell; BM, bone marrow; GSK-3, glycogen synthase kinase 3; ECM, extracellular matrix; NALP3, also called cryopyrin or NLRP3; MAPK, mitogen-activated protein kinase; NF-κB, nuclear factor-kappa B.

**Table 2 ijms-24-03149-t002:** Advantages and disadvantages of animal models using BLM.

Advantages	Disadvantages
The BLM model of PF is the cheapest, easiest, fastest, most reproducible, and most used animal model of IPF	In rodent models, pulmonary fibrosis may regress spontaneously after 28 days
The BLM model is well characterized, clinically relevant, and capable of inducing fibrosis by multiple routes of administration	Attempts have been made to model IPF using different species, routes, and administration methods, but there is no consistency across laboratories
Rodent models are the most useful models for elucidating disease pathogenesis and evaluating preclinical treatments	Administration of compounds tested prophylactically is initiated before fibrosis is induced; when tested therapeutically, it is initiated after fibrosis is established
The current state of preclinical trials can be improved by addressing issues such as time course of treatment, animal size and characteristics, clinically relevant treatment endpoints, and reproducibility of treatment results	Animal models must be carefully selected, designed, and implemented to bridge the translational gap between benchside and bedside

## Data Availability

Not applicable.

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
