# Peer review of "Immune Mechanisms of Pulmonary Fibrosis with Bleomycin"

_ijms, 2023, doi:10.3390/ijms24043149_

Round 1

Reviewer 1 Report

The review written by Ishida Y. et al, focuses on PF pathogenesis in an animal model of BLM-induced PF. The review is overall well-written and facilitates understanding for readers.

The reviewer raises several points to be revised before publication.

1) Title: The authors focus on cytokines, chemokines, growth factors, and matrix feeding. In fact, many factors and mechanisms other than them also contribute to PF pathogenesis. The word “Molecular” is broader application for this review. The word “Immune” instead of “Molecular” is maybe better.

2) Table 1: The authors listed many studies of BLM-induced PF, however, it is not all studies. The title “Studies of BLM-induced PF model” should be changed to “Studies of BLM-induced PF model focused on immune mechanism” etc.

3) The BLM model is the most widely used and best-characterized mouse model, but it has significant limitation in several points. It is recommended that the authors should list up advantages and disadvantages of BLM model in comparison to human IPF as one table.

4) Both fibroblasts and myofibroblast mainly contributes to Phase III in BLM-induced PF as well as human IPF. Both fibroblasts and myofibroblast should be reviewed more in detail (ex: 3.3.4. Fibroblasts, 3.3.5. Myofibroblasts).

5) Recent many studies of BLM model demonstrate that especially macrophages and fibroblast are divided into several cell types using single cell RNA-seq, which made a big impact in PF research. The review seems to lack this point. Additionally, detail cell types in these cells should be described using some references in this review.

6) Line 456: “TGF-b” is incorrect.

Author Response

Reviewer 1

1) Title: The authors focus on cytokines, chemokines, growth factors, and matrix feeding. In fact, many factors and mechanisms other than them also contribute to PF pathogenesis. The word “Molecular” is broader application for this review. The word “Immune” instead of “Molecular” is maybe better.

As the reviewer advised, we changed the title to "Immune mechanisms of pulmonary fibrosis with bleomycin" (line 2).

2) Table 1: The authors listed many studies of BLM-induced PF, however, it is not all studies. The title “Studies of BLM-induced PF model” should be changed to “Studies of BLM-induced PF model focused on immune mechanism” etc.

As the reviewer advised, we changed the title of Table 1 to " Studies of BLM-induced PF model focused on immune mechanism" (line 87).

3) The BLM model is the most widely used and best-characterized mouse model, but it has significant limitation in several points. It is recommended that the authors should list up advantages and disadvantages of BLM model in comparison to human IPF as one table.

On the advice of the reviewer, we added some sentences in the Animal Models section about the differences between animal BLM models and human IPF (line 97-124).

4) Both fibroblasts and myofibroblast mainly contributes to Phase III in BLM-induced PF as well as human IPF. Both fibroblasts and myofibroblast should be reviewed more in detail (ex: 3.3.4. Fibroblasts, 3.3.5. Myofibroblasts).

As the reviewer advised, we added the 3.3.4. Fibroblasts and myofibroblasts section in the Phase III (line 583-623).

5) Recent many studies of BLM model demonstrate that especially macrophages and fibroblast are divided into several cell types using single cell RNA-seq, which made a big impact in PF research. The review seems to lack this point. Additionally, detail cell types in these cells should be described using some references in this review.

On the advice of the reviewers, we added some sentences on single-cell RNA-seq in the Conclusions and future directions (line 675-712).

6) Line 456: “TGF-b” is incorrect.

We modified TGF-b to TGF-β (new line 489).

Reviewer 2 Report

In the present manuscript, the authors reviewed the literatures regarding pathophysiology of wound repairing and remodeling in PF development, especially discusses the role of cytokines, chemokines, growth factors, and matrix feeding in an animal model of BLM-induced PF. The presented information based on references is updated and interesting. However, the review contains some incorrect information. In addition, poor writing makes difficult to read and confused.  

1.    The review needs to be read again thoroughly, there some sentences that don’t make sense (for instance line 328, “CXCL6 is induced in the lungs of patients with IPF after BALF and BLM inhalation…”) and misspellings (for instance line 347, “Macrophage abundance in tissues increases after animals are exposed to various lung toxins, including BLM, suggesting the role of macrophages in acute lung inkury”  

2.    The reference 226 for the line 461, “Secreted modular calcium-binding protein 2 (SMOC2), a factor expressed in almost all tissues, was found to inhibit lung fibrosis progression in SMOC2-/-“ is not described nor discussed for lung fibrosis. In addition, a report by Gerarduzzi et al demonstrated that silencing SMOC2 ameliorates kidney fibrosis by inhibiting fibroblast to myofibroblast transformation (JCI Insight. 2017 Apr 20;2(8):e90299. doi: 10.1172/jci.insight.90299.). The discussion in the review about SMOC2 in FP is confused.

Author Response

Reviewer 2

  1. The review needs to be read again thoroughly, there some sentences that don’t make sense (for instance line 328, “CXCL6 is induced in the lungs of patients with IPF after BALF and BLM inhalation…”) and misspellings (for instance line 347, “Macrophage abundance in tissues increases after animals are exposed to various lung toxins, including BLM, suggesting the role of macrophages in acute lung inkury”

Our manuscript has been edited in English. We modified the line 357-359 as “CXCL6 is enhanced in the BALF of IPF patients. Neutralization with anti-CXCL6 antibody reduced pulmonary neutrophil infiltration, IL-1β, CXCL1, TIMP-1 expression, and collagen deposition in BLM-treated mice (157)”

We also modified the line 374-376 as “Macrophage abundance in tissues increases after animals are exposed to various lung toxins, including BLM (172-174), suggesting that macrophages play a role in acute lung injury.”

  1. The reference 226 for the line 461, “Secreted modular calcium-binding protein 2 (SMOC2), a factor expressed in almost all tissues, was found to inhibit lung fibrosis progression in SMOC2-/-“ is not described nor discussed for lung fibrosis. In addition, a report by Gerarduzzi et al demonstrated that silencing SMOC2 ameliorates kidney fibrosis by inhibiting fibroblast to myofibroblast transformation (JCI Insight. 2017 Apr 20;2(8):e90299. doi: 10.1172/jci.insight.90299.). The discussion in the review about SMOC2 in FP is confused.

Unfortunately, the reference 226 (new 229) was incorrect and has been corrected as “Suppression of SMOC2 reduces bleomycin (BLM)-induced pulmonary fibrosis by inhibition of TGF-β1/SMADs pathway. Biomed Pharmacother. 2018;105:841-847. doi: 10.1016/j.biopha.2018.03.058”.

Reviewer 3 Report

The manuscript entitled " Molecular mechanisms of pulmonary fibrosis with bleomycin" in which the authors examined the common mechanisms of lung wound healing responses after bleomycin-induced lung injury and described the pathogenesis of the most common pulmonary fibrosis. They also demonstrated the role of cytokines, chemokines, growth factors, and matrix feeding in an animal model of bleomycin-induced pulmonary fibrosis.

The work is understandable and the topic is important. The scientific narrative is well structured and flows naturally from one idea to the next.

However, this paper suffers from few shortcomings that if modified would make the manuscript very suitable for publication in International Journal of Molecular Sciences.

Shortcomings:

1-      The authors demonstrate the effects of some treatment options against bleomycin-induced pulmonary fibrosis. However, I think if they write in brief about more therapeutic or preventive options will enrich their review such as MiR-200, Sodium Houttuyfonate, Amifostine Analog, DRDE-30, a RANKL peptide (MHP1-can), irbesartan ……..etc especially the effects of those drugs in experimental pulmonary fibrosis have been shown recently with different molecular mechanisms.

2-      The authors focused on the bleomycin-induced pulmonary fibrosis in vivo in mice models. What about the effect of bleomycin-induced pulmonary fibrosis in rats? are the molecular mechanisms of pathogenesis of pulmonary fibrosis-induced by bleomycin similar?

3-      Please define the abbreviated words in the first mention then write the abbreviation after that in the whole manuscript. For example (FITC in line 46, TGF-β in line 85, COPD, WT mice, IL-1β, TNF-α, PDGF………..etc

4-      Please add all the abbreviated words in table 1 in table legend. Also ECM, and EMT should be defined in figure 1 like EMT: Epithelial-mesenchymal transition,……..etc

Author Response

Reviewer 3

Shortcomings:

1-      The authors demonstrate the effects of some treatment options against bleomycin-induced pulmonary fibrosis. However, I think if they write in brief about more therapeutic or preventive options will enrich their review such as MiR-200, Sodium Houttuyfonate, Amifostine Analog, DRDE-30, a RANKL peptide (MHP1-can), irbesartan ……..etc especially the effects of those drugs in experimental pulmonary fibrosis have been shown recently with different molecular mechanisms.

On the advice of the reviewers, we have added text in the Conclusions and future directions (line 698-712) regarding treatment options for pulmonary fibrosis with bleomycin.

2-      The authors focused on the bleomycin-induced pulmonary fibrosis in vivo in mice models. What about the effect of bleomycin-induced pulmonary fibrosis in rats? are the molecular mechanisms of pathogenesis of pulmonary fibrosis-induced by bleomycin similar?

Rodent models, including rats, show similar pathophysiology of bleomycin-induced pulmonary fibrosis. The number of studies of bleomycin-induced pulmonary fibrosis using rats is very small, but we have included them in the text (line 434-439, 500-508, 618-623).

3-      Please define the abbreviated words in the first mention then write the abbreviation after that in the whole manuscript. For example (FITC in line 46, TGF-β in line 85, COPD, WT mice, IL-1β, TNF-α, PDGF………..etc

We have corrected the following.

Line 46, fluorescein isothiocyanate (FITC)

Line 85, transforming growth factor (TGF)-β

Line 152-153, chronic obstructive pulmonary disease (COPD)

Line 259, wild-type (WT) mice

Line 203, interleukin (IL)-13

Line 252, interleukin (IL)-1α

Line 225-226, platelet-derived growth factor (PDGF); tumor necrosis factor (TNF)-α

4-      Please add all the abbreviated words in table 1 in table legend. Also ECM, and EMT should be defined in figure 1 like EMT: Epithelial-mesenchymal transition,……..etc

We would like to add a legend to Table 1 as follows,

Abbreviations: Mmp, matrix metalloproteinase; EMT, epithelial-mesenchymal transition; IL, interleukin; MIP-1α, macrophage inflammatory protein-1α; BAL, bronchoalveolar lavage; IGFBP3, insulin-like growth factor binding protein 3; WT, wild-type; PTGS2, prostaglandin–endoperoxide synthase 2; MyD88, myeloid differentiation factor 88; IL-1R1, IL-1 receptor 1; TGF-β, transforming growth factor-β; MSC, mesenchymal stem cell; hfPMSC, human placental MSCs of fetal origin; BALF, BAL fluid; PF, pulmonary fibrosis; IL-17RA, IL-17 receptor; ST2, chain suppression of tumorigenicity 2; CCL, CC chemokine ligand; CCR, CC chemokine receptor; TNF, tumor necrosis factor; CXCR, CXC chemokine receptor; CXCL, CXC chemokine ligand; Col I, collagen type I; CX3CR, CX3C chemokine receptor; PI3K, phosphatidylinositol 3-kinase; mTOR, mammalian target of rapamycin; JAK, Janus kinase; p-STAT, phosphorylated-signal transducer and activator of transcription; DC, dendritic cell; FGF, fibroblast growth factor; Treg, regulatory T-cell; PDGF, platelet-derived growth factor; bFGF, basic fibroblast growth factor; VEGF, vascular endothelial growth factor; CTGF, connective tissue growth factor; scFv, single-chain variable fragment antibody; IGF-I, insulin-like growth factor I; iPS cell, induced pluripotent stem cell; BM, bone marrow; GSK-3, glycogen synthase kinase 3; ECM, extracellular matrix; NALP3, also called cryopyrin or NLRP3; MAPK, mitogen-activated protein kinase; NF-κB, nuclear factor-kappa B

We have corrected the following.

Line 136, extracellular matrix (ECM)

Line 137-138, epithelial-mesenchymal transition (EMT)

Round 2

Reviewer 1 Report

The revised manuscript was well improved except the following 2 points.

It is necessary for the authors to edit them before publication.

1) Line 97-124; As the reviewer mentioned in previous comment, the differences between animal BLM models and human IPF (advantages and disadvantages) should be indicated as one table so that readers can understand the point more clearly.

2) Reference 299 (Line 1535): The description may be wrong, please check it again.

Author Response

1) Line 97-124; As the reviewer mentioned in previous comment, the differences between animal BLM models and human IPF (advantages and disadvantages) should be indicated as one table so that readers can understand the point more clearly.

On the advice of the reviewers, we have added Table 2 (line 122 and 1596-1597).

2) Reference 299 (Line 1535): The description may be wrong, please check it again.

We corrected description of Reference 299 (line 1535-1534).

Reviewer 2 Report

The review needs to be revised thoroughly especially the second half. The sentences that don’t make sense were not limited to the sentences in the comments. The authors only revised few sentences that mentioned in the comments.

Author Response

The review needs to be revised thoroughly especially the second half. The sentences that don’t make sense were not limited to the sentences in the comments. The authors only revised few sentences that mentioned in the comments.

On the advice of the reviewers, the manuscript was carefully edited in English by an English speaker (please see the attached certification).
